# Development of CO_2_-Selective Polyimide-Based Gas Separation Membranes Using Crown Ether and Polydimethylsiloxane

**DOI:** 10.3390/polym13121927

**Published:** 2021-06-10

**Authors:** Dongyoung Kim, Iqubal Hossain, Asmaul Husna, Tae-Hyun Kim

**Affiliations:** 1Organic Material Synthesis Laboratory, Department of Chemistry, Incheon National University, 119 Academy-ro, Yeonsu-gu, Incheon 22012, Korea; bluegod94@nate.com (D.K.); iqubal.chem.ru.08@gmail.com (I.H.); shopnilchem@inu.ac.kr (A.H.); 2Research Institute of Basic Sciences, Incheon National University, 119 Academy-ro, Yeonsu-gu, Incheon 22012, Korea

**Keywords:** copolymer synthesis, crown ether, CO_2_ separation, structure–property relationship

## Abstract

A series of CO_2_-selective polyimides (CE-PDMS-PI-x) was synthesized by copolymerizing crown ether diamine (trans-diamino-DB18C6) and PDMS-diamine with 4,4′-(hexafluoroisopropylidene) di-phthalic anhydride (6FDA) through the polycondensation reaction. The structural characteristics of the copolymers and corresponding membranes were characterized by nuclear magnetic resonance (NMR), infrared spectroscopy (ATR-FTIR), thermogravimetric analysis (TGA), differential scanning calorimetry (DSC), X-ray diffraction (XRD), and gel permeation chromatography (GPC). The effect of PDMS loading on the CE-PDMS-PI-x copolymers was further analyzed and a very good structure–property relationship was found. A well-distributed soft PDMS unit played a key role in the membrane’s morphology, in which improved CO_2_-separation performance was observed at a low PDMS content (5 wt %). In contrast, the fine-grained phase separation adversely affected the separation behavior at a certain level of PDMS loading, and the PDMS was found to provide a flexible gas-diffusion path, affecting only the permeability without changing the selective gas-separation performance for the copolymers with a PDMS content of 20% or above.

## 1. Introduction

Fossil fuels are widely used as primary energy sources to meet the exponentially increasing demands for energy, which are being driven by population expansion [1,2]. However, this level of consumption has led to excessive greenhouse gas emissions, resulting in unprecedented climate and environmental changes, along with a rise in the average temperature of the Earth’s atmosphere [3]. Carbon dioxide (CO_2_) is mainly generated by human activities—such as industry, power generation, and transportation—and it accounts for the largest portion of the gases that contribute to the greenhouse effect and global warming [4,5]. Unfortunately, demand for fossil fuels is still high, and they are likely to remain the world’s primary energy source until economically favorable alternatives are developed and reliably commercialized to replace them. CO_2_ emissions will inevitably continue to increase until alternative energy sources and technologies are found [3,4].

Carbon capture and storage (CCS) technologies can be used to capture, compress, and transport CO_2_ [1,2]. This CO_2_ can be stored on the seafloor or underground, for example. These technologies are also related to carbon capture utilization and storage, which involves converting the CO_2_ into methanol or polymers to be recycled as higher value-added resources. Since CO_2_ separation and storage technologies are essential for implementing long-term strategies to conserve the Earth’s environment, much research has focused on improving existing methods and proposing new approaches [1,2,3,4]. One of these, the membrane-based separation method, differs from the others in that it does not involve any phase transformations and is thus a regeneration process. This feature makes the method more energy efficient than those based on equilibrium reactions, such as absorption and adsorption processes [1,2,3,4,5,6,7,8] This technology is also eco-friendly because the maintenance cost is relatively low, the process is simple and modularized, and the emission of harmful substances is reduced [8,9].

While a wide range of polymer-based membranes has been studied and developed, only a few have been commercialized for gas-separation applications. This discrepancy is attributed to unresolved obstacles to the commercialization of the developed materials. Specifically, there is a trade-off for most membranes between permeability and selectivity [10,11]. This means that implementing a membrane with superior performance requires the development of materials that can simultaneously improve both permeability and selectivity.

Polyimide is one of the most widely used glassy polymers, and it provides excellent thermal stability, chemical durability, and mechanical strength [8,9,12,13,14,15]. For that reason, the material has been widely used in a variety of applications, including batteries, shape memory devices, sensors, wearable devices, and gas-separation membranes [13]. Polyimide can be synthesized via the condensation polymerization of diamine and dianhydride, and numerous polymer combinations with various shapes can be achieved by varying its monomer structure. As one of the most widely used glassy polymers, the material provides superior thermal and mechanical stability, moderate selectivity to gases, and relatively low permeability. Accordingly, to use the material as a gas-separation membrane, it is necessary to design an appropriate structure. One possible method to achieve this would be to copolymerize the material with monomers that provide high permeability.

Copolymerization is a polymer-synthesis method that integrates substances with different properties into a single polymeric material [8,15,16]. With this technique, it is possible to modify or improve the properties of a homopolymer composed of a single monomer unit. For example, copolymers using polydimethylsiloxane (PDMS) have been studied to improve the low permeability of glassy polymers. Gurr et al. [17] fabricated a thin-film composite membrane (TFC) using PDMS-terminated polyimide via click chemistry. They reported that the fabricated membrane provided excellent coating characteristics due to the superior properties of the polyimide, and, at the same time, its permeability was increased by about four-fold with increased PDMS chain length. Park et al. [18] subjected PDMS-containing polyimide to ultraviolet (UV) rays in order to form crosslinks, and examined and compared the gas-permeability characteristics of the resulting material. The addition of PDMS was found to increase the free volume, improving the permeability, but also reduced the selectivity of the fabricated membrane. However, after the crosslinks formed, the free volume decreased, thereby improving the selectivity [18]. These studies indicated that adding an excessive amount of PDMS to improve permeability may significantly reduce selectivity; it is therefore important to ensure that an appropriate amount of PDMS is used.

Crown ether is a ring-shaped compound that contains ether bonds (C-O-C). The cavities inside its rings can trap molecules, metal ions, or other substances [19]. The trapping capacity depends on the cavity size. Because of this property, crown ether is expected to facilitate the selective separation of gases if the size of its cavities is comparable or equivalent to the kinetic size of the target gases. In particular, 18-crown-6 is reported to have a cavity size equivalent to the kinetic diameter of CO_2_, which is 3.3 Å [20,21,22]. The ether bonds present in the material may further improve the affinity with CO_2_. Furthermore, the ether bonds are more rigid and stiff than linear ether bonds, such as those observed in polyethylene(glycol) (PEG) [21,22]. This material is also expected to be able to compensate for the loss in size selectivity arising from the use of flexible materials.

The present study aimed to synthesize a polymer material with improved permeability, higher affinity with CO_2_, and excellent selectivity by copolymerizing PDMS, a rubbery polymer with a low glass-transition temperature (*T_g_*), and DB18C6, a rigid crown ether, to improve the low permeability of the polyimide. The developed material was then formed into a film shape, and its applicability as a CO_2_ separation membrane was evaluated.

## 2. Materials and Methods

### 2.1. Materials

4,4′-(hexafluoroisopropylidene) di-phthalic anhydride (6FDA) and dibenzo-18-crown-6 (DB18C6) were purchased from TCI Co. Ltd. (Seoul, Korea). and were used as obtained. Bis(3-aminopropyl) terminated PDMS (*M_n_* = 2500), triethylamine, hydrazine monohydrate, acetic anhydride, nitric acid, and anhydrous acetic acid were obtained from Sigma-Aldrich Korea (Yongin, Korea) and were used without purification. Pd/C catalyst (palladium, 10% on carbon, Type 487, dry) was purchased from Alfa-Aesar (Seoul, Korea). Ethanol, isopropanol, tetrahydrofuran, dimethylacetamide, dimethylsulfoxide, and dimethylformamide were purchased from DaeJung Chemicals & Metals Co. Ltd. (Shiheung, Korea). *Trans*-diamino-DB18C6 (3) was synthesized following previous reports [21,22], and the procedures are described in the Appendix A. All other chemicals were obtained from commercial sources.

### 2.2. Synthesis of the CO_2_-Selective CE-PDMS-x Polymer (4)

A two-step synthesis (that is, polyamic-acid preparation followed by imidization) was carried out to prepare the CE-PDMS-x copolyimides with six different compositions following the literature [9,15].

Synthesis of the CE-PDMS-5: The general procedure for the synthesis of the CE-PDMS-x polymers is as follows: trans-diamino-DB18C6(3.3) (200 mg, 5.12 × 10^−4^ mol) was first dissolved in DMAc (2 mL) in an oven-dried two-neck round bottom flask equipped with a reflux condenser, nitrogen inlet, and magnetic stirrer bar. Once the monomer was completely dissolved, 4 mL of THF was added to the solution with PDMS (10 mg, 4 × 10^−6^ mol) (the solvent ratio is DMAc:THF = 1:2 (v/v%)). 6FDA (229.34 mg, 5.16 × 10^−4^ mol) was poured into this solution, and it was stirred for 48 h under N_2_ atmosphere. After that, the reaction mixture was heated to 50 °C, followed by the addition of triethylamine (209 mg, 2.07 × 10^−3^ mol) and acetic anhydride (209 mg, 2.07 × 10^−3^ mol). Then, the solution was vigorously stirred and refluxed for 4 h at 110 °C to complete the imidization. The viscous mixture was then cooled to room temperature and precipitated in isopropanol. The resulting yellowish product was washed with ethanol and methanol. The product was dried under a vacuum oven for 24 h at 80 °C. If reprecipitation was required, the same process was repeated after dissolution in chloroform. δ_H_(400 MHz, CDCl_3_) 8.1–7.72(6 H, m, H_a,b,c_), 7.05–6.80(5.76 H, m, H_d,e,f_), 4.32–3.80(15.68 H, br, H_g,h_), 0.14–0.01(1.82 H, br, H_i_)



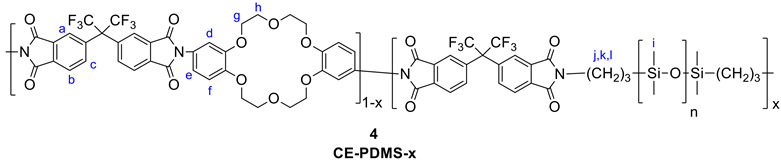



All other copolymers with the same general structure, CE-PDMS-x (4), with different loadings of PDMS were prepared using the same method as CE-PDMS-5 above with different monomer ratios of CE and PDMS. Their NMR characteristics are presented below:

CE-PDMS-5: δH(400 MHz, CDCl3) 8.1–7.74(6 H, m, Ha,b,c), 7.07–6.84(5.59 H, m, Hd,e,f), 4.35–3.84(15.33 H, br, Hg,h), 0.15–0.01(1.97 H, br, Hi)

CE-PDMS-10: δH(400 MHz, CDCl3) 8.08–7.80(6 H, m, Ha,b,c), 7.04–6.84(5.92 H, m, Hd,e,f), 4.33–3.91(15.86 H, br, Hg,h), 0.14–0.01(5.91 H, br, Hi)

CE-PDMS-20: δH(400 MHz, CDCl3) 8.04–7.76(6 H, m, Ha,b,c), 6.97–6.75(5.72 H, m, Hd,e,f), 4.33–3.82(15.22 H, br, Hg,h), 3.61(1.32 H, t, J = 8 Hz, Hj), 1.63(1.67 H, m, Hk), 0.50(1.42H, t, J = 8 Hz, Hl), 0.11–0.10(45.53 H, br, Hi)

CE-PDMS-30: δH(400 MHz, CDCl3) 8.00–7.77(6 H, m, Ha,b,c), 6.96–6.75(5.46 H, m, Hd,e,f), 4.25–3.85(14.06 H, br, Hg,h), 3.61(1.99 H, t, J = 8 Hz, Hj), 1.63(2.32 H, m, Hk), 0.50(2.21 H, t, J = 8Hz, Hl), 0.10–0.07(45.53 H, br, Hi)

CE-PDMS-40: δH(400 MHz, CDCl3) 8.00–7.77(6 H, m, Ha,b,c), 6.97–6.76(5.47 H, m, Hd,e,f), 4.25–3.83(13.47 H, br, Hg,h), 3.61(2.74 H, t, J = 8 Hz, Hj), 1.63(2.91 H, m, Hk), 0.50(2.97H, t, J = 8Hz, Hl), 0.12–0.11(45.53 H, br, Hi)

Synthesis of 6FDA-CE PI: PDMS was not added during the synthesis of this homopolymer (6FDA-CE(5)). The method was otherwise the same as that used for CE-PDMS-5, but only DMAc solvent was used in the reaction. δH(400 MHz, CDCl_3_) 8.1–7.72(6 H, m, Ha,b,c), 7.05–6.80(6 H, m, Hd,e,f), 4.40–3.75(16 H, br, Hg,h)



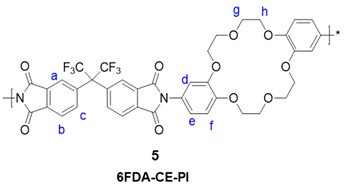



### 2.3. Membrane Fabrication for CE-PDMS-x

All membranes were prepared in an *N,N*-dimethylacetamide (DMAc) solution of the corresponding polymers, and the solution-casting method was employed. CE-PDMS-x polymers (**4**) (150 mg) were dissolved in 8 mL of DMAc and stirred overnight to obtain a homogeneous solution. The polymer solution was then filtered using a cotton plug and poured into a polypropylene petri dish. The filtered solution was dried under vacuum at 40 °C for 8 h followed by drying at 60 °C overnight to evaporate the solvent completely. The membranes were peeled off carefully, and the resultant membrane was kept in a vacuum oven at 80 °C for 24 h to remove the residual solvent.

Characterization of the polymer membranes followed our previous report [6], and is described in the Appendix A.

## 3. Results and Discussion

### 3.1. Synthesis and Characterization of Polymers

Polyimide was synthesized via the condensation polymerization of diamine and dianhydride; therefore, amine functional groups (−NH_2_) were required to introduce DB18C6 into the polyimide structure. DB18C6 (**1**) was subjected to nitration to add amine functional groups to the material. The nitro functional groups (−NO_2_) were then reduced using Pd/C catalysts and N_2_H_4_ as a reducing agent (Appendix A). The structures of the synthesized monomers dinitro-DB18C6 (**2**) and diamino-DB18C6 (**3**) were analyzed using proton nuclear magnetic resonance (^1^H NMR) and Fourier transform infrared-attenuated total reflectance (ATR-FTIR). After the reduction process, shown in Appendix A, ^1^H NMR data showed the appearance of *H_f_*, a new peak corresponding to the amine functional groups (**3**) (Appendix A). In the ATR-FTIR spectra, two peaks were observed at 3424 cm^−1^ and 3356 cm^−1^, which were the N-H stretching absorption peaks resulting from the primary amine functional groups (Appendix A) [21,23]. These results confirmed the successful synthesis of the intended monomers (**3**) for use in further polyimide synthesis.

6FDA, PDMS, and diamino-DB18C6 (**3**) were used as the comonomers to fabricate the crown ether and CE-PDMS-based polyimide coatings (PIs) with different amounts of CE and PDMS (**4**) (Scheme 1). DMAc, which is widely used for polyimide synthesis, was employed as a solvent, together with THF as a cosolvent to dissolve the PDMS. Subsequently, chemical imidization was applied to synthesize the CE-PDMS polyimides (**4**) using triethtyl amine (TEA) and acetic anhydride. These polyimides were designed with varying ratios of monomers, summarized in Table 1, so that the effect of PDMS content on permeability performance could be examined. The polyimide was also prepared with only 6FDA and diamino-DB18C6 (**3**) to produce 6FDA-crown ether-based polyimide (6FDA-CE-PI), which was used as a reference. The PDMS content was adjusted from low (2.5, 5, 10 wt %) to moderately high (20, 30, 40 wt %) relative to compound **3**, and the samples were referred to as CE-PDMS-2.5, CE-PDMS-5, CE-PDMS-10, CE-PDMS-20, CE-PDMS-30, and CE-PDMS-40, respectively. When the amount of PDMS was higher than 40 wt%, the polymer failed to yield a self-standing membrane.

The structure of the synthesized polymers was analyzed using ^1^H NMR and ATR-FTIR. In 6FDA-CE-PI, CE-PDMS-2.5, CE-PDMS-5, and CE-PDMS-10, which had low PDMS contents, a peak corresponding to the hydrogens of the benzene rings of 6FDA was observed at 8.07–7.80 ppm, along with a peak corresponding to the hydrogens of the benzene rings of the crown ether at 7.05–6.84 ppm (Figure 1a). The peak observed at 4.30–3.90 ppm was attributed to the hydrogens in the ether bonds, while the peak at 0.12–0.03 ppm resulted from the -CH_3_ groups of the PDMS. These results revealed the structure of the synthesized polymers. The compositional ratio of the crown ether to PDMS in each polymer was determined based on the relative integral ratio of the hydrogens contained in the ether bonds of the former to the hydrogens present in the repeating units of the latter. In the FTIR spectra, peaks were observed from the vibration of the characteristic bonds of each polymer. Absorption peaks at 1785 cm^−1^, 1723 cm^−1^, and 721 cm^−1^ were attributed to C=O asymmetric stretching, symmetric stretching, and deformation of the imide bonds, respectively (Appendix A) [8,9,15]. The peaks indicating the presence of the monomers used are shown in Figure 1b. The peaks at 1020 cm^−1^ and 800 cm^−1^ in particular were found to increase in intensity with increasing PDMS content.

Meanwhile, CE-PDMS-20, CE-PDMS-30, and CE-PDMS-40, which had relatively high PDMS content, produced ^1^H NMR and ATR-FTIR spectra with shapes similar to those with lower PDMS content (Appendix A). Increased amounts of PDMS were found to increase the intensity of *H_i_* in the NMR spectra, and, at the same time, *H_j_*, *H_k_*, and *H_l_* were also observed.

In the FTIR spectra, high-intensity absorption peaks corresponding to PDMS appeared at the wavenumbers 1260 cm^−1^ (S-CH_3_ bending), 1020 cm^−1^ (Si-O-Si sym. stretching), and 800 cm^−1^ (Si-CH_3_ rocking) (Appendix A). However, in the NMR spectra, two new peaks appeared in the CE-PDMS-x copolymers (with high loading of PDMS over 20 wt %) at 7.65 ppm and 7.75 ppm corresponding to the phenyl proton of 6FDA-PDMS (Appendix A). These were attributed to the phenyl protons of the 6FDA-PDMS segments, which were not clearly separated at a low loading of PDMS.

The polymers with higher PDMS content exhibited a larger polydispersity index (PDI) (Table 1). It is likely that either the crown ether and PDMS did not react with 6FDA due to the large difference in solubility or that the PDMS, which is more flexible, was copolymerized while being chelated by the crown ether [24].

The polymers synthesized with different compositions were fabricated into membranes via solution-casting using DMAc as a solvent. Vacuum drying was then applied to obtain non-brittle, defect-free, and self-standing membranes (Appendix A). The stickiness of the membranes increased with the PDMS content due to its flexible properties.

### 3.2. Thermal Properties of the CE-PDMS-x Polymers

Initially, the thermal properties of the polymers with low PDMS content—including 6FDA-CE-PI, CE-PDMS-1, CE-PDMS-2, and CE-PDMS-3—were measured using thermogravimetric analysis (TGA) and analyzed (Figure 2). The 6FDA-CE-PI homopolymer exhibited a two-step weight loss (Figure 2a). The first degradation was found at around 450 °C, which was attributed to the weight loss caused by the decomposition of the crown ether [21]. This was followed by a weight loss at above 500 °C caused by the decomposition of the main chains of the polyimide [8,13,15]. In the polymers copolymerized with PDMS, the degradation temperature of the crown ether tended to decrease as the flexible PDMS content increased, as did its ratio. The degradation temperature of the PDMS was slightly lower than that of the main chains of polyimide at 480 °C. The degradation continued, along with that of the main chains of the polyimide, until the temperature reached 600 °C, as confirmed by the deliberative curves (Figure 2b). The char remaining at up to 800 °C increased with the PDMS content and showed excellent thermal stability. Overall, these membranes showed excellent thermal properties, as required for gas-separation applications.

The polymers with moderately high PDMS content—including CE-PDMS-20, CE-PDMS-30, and CE-PDMS-40—exhibited different weight losses with temperature than those with lower PDMS content (Appendix A). When the PDMS content was low, the polymers exhibited a two-step degradation pattern; however, as the PDMS content increased, the polymers started to show different behavior. CE-PDMS-30 in particular, which had a high PDI value, exhibited a three-step degradation pattern. This indicated that some low-molecular-weight polyimide was formed as intended, due to the difference in solubility of the two monomers.

Next, the *T_g_* of the synthesized polymers was measured using differential scanning calorimetry (DSC) (Figure 3). The *T_g_* of 6FDA-CE PI was measured as 222 °C, which was lower than that of typical polyimide materials. This was attributed to the presence of several ether bonds in the main chains of the polymer, despite the higher rigidity provided by the crown ether [21]. The polymers with lower PDMS content were expected to have a lower *T_g_* with increased flexible PDMS content; however, the *T_g_* was found to increase continuously.

A study of the thermal properties of polyimide materials containing PDMS showed that the *T_g_* was higher when the PDMS content was lower; this was because the hard polyimide and soft PDMS were more uniformly mixed under these conditions so that aggregation of the latter could not occur [25]. This implied that, during the copolymerization process, the PDMS could be more uniformly distributed and thus ensure better performance at the specific level of content. In our previous study, the *T_g_* was found to increase with the amount of PDMS present inside the main chains of the polyimide, but started to decrease due to aggregation [26].

A study of the thermal properties of polyimide materials containing PDMS showed that the *T_g_* was higher when the PDMS content was lower; this was because the hard polyimide and soft PDMS were more uniformly mixed under these conditions so that aggregation of the latter could not occur [25]. This implied that during the copolymerization process, the PDMS could be more uniformly distributed and thus ensure better performance at the specific level of content. In our previous study, the *T_g_* was found to increase with the amount of PDMS present inside the main chains of the polyimide, but started to decrease due to aggregation [26].

However, a different DSC curve was observed at a high loading of PDMS (Appendix A)**,** where the *T_g_* value of all the copolymer membranes was dramatically reduced. This may be attributed to the high amount of soft PDMS present in the copolymer structure, resulting in very soft copolymer membranes; therefore, the gas separation property in these membranes will be dominated by the soft PDMS content. A similar DSC result has been reported for poly(imide siloxane) copolymer membranes by Lee et al. [25].

Phase-separation behavior should be examined to provide further evidence. To this end, the surface characteristics of the fabricated membranes were studied.

### 3.3. Morphology of CE-PDMS-x Membranes Determined Using AFM and Wide-Angle X-ray Diffraction (WAXD) Analysis

The effect of PDMS content on the phase-separation behavior of the membranes was examined by analyzing the morphological characteristics of their surfaces using atomic force microscopy (AFM) (Figure 4). Many studies have reported that phase separation occurs in polyimides with polysiloxane units between the hard and soft segments due to low mixing entropy [27,28]. 6FDA-CE PI with no PDMS generally showed a uniform morphology. However, the polymers that copolymerized with PDMS—including CE-PDMS-2.5, CE-PDMS-5, and CE-PDMS-10—were found to have dark regions on the surface, which corresponded to PDMS (Figure 4). This indicated that phase separation had occurred.

The phase-separation behavior caused by PDMS varied depending on its content. In CE-PDMS-2.5, the darker region corresponding to PDMS was formed over a long distance, whereas in CE-PDMS-5, the darker regions were uniformly distributed throughout the entire membrane surface. In CE-PDMS-10, the darker regions were well-developed and pronounced rather than being well-distributed; this indicated the occurrence of PDMS aggregation. The ability to achieve uniform phase separation was expected to affect the gas-permeability performance of membranes. However, AFM analysis was not possible using the copolymer membranes with high PDMS content (above 10 wt %) due to their soft nature.

The microphase-separated structures of the membranes were confirmed by transmission electron microscopy observations, as shown in Figure 5. In this TEM analysis, it was not necessary to stain the sample due to the significant difference in the electron density of the inorganic polydimethylsiloxane segments and the organic 6FDA-CE-PI segments. The 6FDA-CE-PI membrane showed a homogeneous and uniform morphology (Figure 5a). However, all the copolymer membranes displayed a phase-separated heterogeneous morphology, where spherical spots corresponding to the PDMS were distributed into the matrix polymer. The concentration of these spots increases with increased PDMS loading. For example, very few spots were present in the CE-PDMS-2.5 membranes at 2.5 wt % PDMS loading (Figure 5b), but an increased number of spheres with uniform distribution was present at 5 wt % PDMS loading (Figure 5c). However, a large aggregation was observed in the CE-PDMS-10 (Figure 5d). The same type of morphology has been reported in other studies which used TEM analysis [25,28].

At high loadings of PDMS (20–40 wt %), the siloxane regions are characterized by higher electron densities more resistant to the transmittance of electrons and, therefore, appear darker in the TEM images (Appendix A). The CE-PDMS-20 membrane shows dark spheres distributed in the matrix in a continuous manner (Appendix A). However, the PDMS phase becomes increasingly well connected at high loadings for CE-PDMS-30 and CE-PDMS-40 copolymer membranes (Appendix A). This morphology indicates that the gas separation property will be controlled and dominated by the well-connected soft PDMS phase in these copolymer membranes.

WAXD measurements were analyzed to investigate the effect of PDMS loading on the chain packing in the CE-PDMS-x copolymer membranes 1, as represented in Figure 6. All the copolymer membranes showed broad peaks, confirming their amorphous nature. However, the peak positions of the curve were regulated by the PDMS loading in the corresponding copolymer membranes. For example, a broad peak located at 2θ = 16.9° was observed in the spectra for the 6FDA-CE-PI membrane, which represented a d-spacing of 5.25 Å, according to Bragg’s law, d = λ/2sin θ, where λ is the wavelength, 1.789 Å, and θ is the scattering angle. This d-spacing was attributed to the average interchain distance of the homopolymer.

As can be seen in Figure 6a, the peak positions of the amorphous halos shifted to higher d-spacings as the PDMS loadings increased up to 5 wt % (CE-PDMS-5). Specifically, the peaks shifted to 5.72 Å and 5.97 Å for CE-PDMS-2.5 and CE-PDMS-5 membranes, respectively, due to the addition of 2.5 and 5 wt % PDMS in the copolymer chain. This indicated that the free volume of these copolymers (PDMS-2.5 and CE-PDMS-5) increased significantly with the incorporation of PDMS. Notably, the amorphous peak of the CE-PDMS-5 copolymer membrane broadened immensely with a 5 wt % loading of PDMS, because the 6FDA-CE-PI and PDMS peaks overlapped. These results indicated that the soft PDMS polymer was randomly mixed/distributed in the rigid 6FDA-CE PI phase without noticeable phase separation. However, two broad peaks corresponding to the 6FDA-CE-PI and PDMS segments were observed in the CE-PDMS-10 copolymer membrane, demonstrating that these phases were separated at a 10% loading due to their different solubilities.

In contrast, different WAXD curves were observed at high PDMS contents (i.e., over 20 wt % loading), with only a single diffraction peak corresponding to the PDMS segment being observed for the three copolymers CE-PDMS-20, CE-PDMS-30, and CE-PDMS-40 (Figure 6b). This result indicated that these copolymer morphologies were completely dominated by the soft phase PDMS and the diffraction pattern for the amorphous PIs phase was non-detectable. However, the free volume depended on the PDMS content and increased with the PDMS loading. Therefore, d-spacing followed the same order as the free volume with PDMS loading, as seen in Figure 6b.

Overall, the WAXD results implied that the free volume in the copolymer membranes increased with PDMS loading up to 5%, and the two different components were well distributed in the copolymer membrane without phase separation. However, the copolymer membrane with 10% PDMS loading, CE-PDMS-10, formed a heterogeneous phase where the PDMS segments aggregated in the rigid polyimide phase. The gas-separation performance using these copolymer membranes was also expected to show the same trend: the gas permeability was predicted to increase with the PDMS loading up to the level at which the soft phase was homogeneously distributed in the rigid PI phase. At higher loadings, the gas permeability was expected to behave differently due to the polymer membranes with aggregated morphology. Similar morphology versus gas-separation performance has been reported elsewhere [26].

### 3.4. Gas-Separation Properties

The gas-separation performance of the CE-PDMS-x copolymer membranes and the effect of PDMS loading on their performance were examined by measuring the pure gas permeabilities of each membrane against N_2_, CH_4_, and CO_2_ at 2 atm and 30 °C.

The gas permeabilities of copolymer films with up to 10 wt% of PDMS followed the order P(CO_2_) > P(N_2_) > P(CH_4_), which was consistent with the size (kinetic diameter) of the gases (Table 2). This trend was typical of rigid polymers, such as copolymers with low PDMS content, which largely rely on molecular sieving for gas transport [9,12,13,18,29]. For example, the 6FDA-CE-PI homopolymer membrane with no PDMS exhibited low gas permeabilities of 3.26, 0.11, and 0.07 Barrer for CO_2_, N_2,_ and CH_4_, respectively. This low gas permeability was attributed to the rigid structure of the crown ether-based PI, making it more difficult for gas molecules to diffuse into and pass through the main chains of the polymer. However, this polymer displayed very high CO_2_ selectivities of 29.6 (CO_2_/N_2_) and 46.6 (CO_2_/CH_4_) over the non-polar gases (N_2_ and CH_4_) with large sizes. This high CO_2_-selectivity could be explained by a combination of the rigid and compact structure of the polyimide with low free volume and the size of the cavities present in DB18C6 that are equivalent to the kinetic size of CO_2_. These allowed the CO_2_ molecules to selectively diffuse (preferentially over the larger gases such as N_2_ and CH_4_). In addition, the overall solubility could be increased by the ether bonds, because they can induce the quadrupole–dipole interaction with CO_2_ and hence increase its solubility selectivity over non-polar gases such as N_2_ and CH_4_.

When the gas-separation performances of the various copolymer membranes, CE-PDMS-x, were compared, the gas permeability in all cases increased gradually up to 5 wt % PDMS loading (Table 2 and Figure 7); this was followed by an unexpected reduction of permeability at 10 wt % (CE-PDMS-10). This increase in gas permeabilities was attributed to an increase in the chain flexibility of the copolymer due to the incorporation of PDMS, which affected the chain packing of the polymer matrix. For example, the CO_2_ permeability of 2.5 and 5 wt % PDMS-loaded copolymer membranes increased by 382% and 441% compared to the pristine PI membrane (Table 2). This enhancement was seen up to a certain level of PDMS loading (here, 5 wt %) that could be distributed well in the rigid polyimide matrix. Aggregated PDMS did not follow the trend of increasing chain flexibility and gas-separation performance in the resulting copolymer. For example, the permeabilities of all gases were reduced by 100% for the 10% PDMS-loaded copolymer membrane, CE-PDMS-10, compare to the CE-PDMS-5 membrane, which was attributed to the agglomeration of the PDMS phase inside the rigid phase. Similar results have been reported elsewhere [26].

The CO_2_/N_2_ selectivity of the copolymer membrane, CE-PDMS-2.5, initially increased by 41% (41.8 for the CE-PDMS-2.5 membrane versus 29.6 for 6FDA-CE-PI), due to a large increase in CO_2_ permeability compared to N_2_ (Table 2 and Figure 7). Further increased loading of PDMS in the copolymer membrane (up to 10% loading) gradually reduced the CO_2_/N_2_ selectivity compared to the CE-PDMS-2.5 membrane. This was expected because the increase of chain flexibility imparted by PDMS effectively reduced the size selectivity. However, the selectivities of the 5% and 10% PDMS-loaded copolymer membranes, CE-PDMS-5 and CE-PDMS-10, remained higher than that of the 6FDA-CE-PI membrane. Similar patterns of simultaneous improvements in permeability and selectivity to various gas pairs with increased PDMS loading have been reported elsewhere [26,29,30]. In comparison, the CO_2_/CH_4_ selectivity gradually decreased with an increase in PDMS loading (up to 10 wt % loading). However, the CO_2_/CH_4_ selectivity of these copolymer membranes was still relatively high at PDMS loadings up to 10 wt %.

The sequence of gas permeabilities changed when the PDMS content reached 20 wt % to P(CO_2_) > P(CH_4_) > P(N_2_), which was consistent with the rubbery membranes containing PDMS/PEG-type flexible units (Figure 7. The incorporation of rubbery PDMS increased the diffusion of CH_4_ much more than the diffusion of N_2_, and the high condensability and better interaction of CH_4_ with PDMS yielded high CH_4_ solubility. Consequently, the permeability of CH_4_ surpassed the permeability of N_2_ as the PDMS increased in the copolymer membrane.

Notably, the gas permeability of the copolymer membranes showed a typical behavior, with a dramatic increase of many orders of magnitude when the PDMS loading rose from 10% to 20%, and further rapid growth up to 38% loading (Table 2 and Figure 7. A similar behavior of gas separation in copolymer membranes that combine the rigid polymer with PDMS has been reported in many studies [29,31,32]. This result is attributed to the percolation threshold of gas separation at a certain volume fraction of flexible PDMS loading. The permeability of the copolymer membrane containing flexible PDMS units increased slowly below a certain threshold value of PDMS (a volume fraction of 0.1–0.3) and was followed by an exponential increase in permeability above and near the threshold value [31,32]. Above the threshold value of PDMS, the copolymer became an elastomeric phase with high flexibility that allowed gas molecules to diffuse preferentially through the less-resistant PDMS microdomains and rapidly enhanced the gas permeability.

The selectivities of CO_2_/N_2_ and CO_2_/CH_4_ dropped to 10.6 and 3.5 at 20% PDMS loadings and remained the same with further increases (Table 2 and Figure 7). This was because the PDMS phase became the main gas-transport pathway above and near the threshold value, resulting in constant selectivity to the gas pairs regardless of PDMS loadings. Similar results have been reported elsewhere [29].

### 3.5. Permeability Versus Selectivity

To compare the gas-separation performance of the synthesized membranes with those reported in previous studies, the corresponding data points were analyzed using a Robeson plot (Figure 8) [11,29,30,33,34,35,36,37,38]. Although our polymer membranes did not show performance crossing the Robeson upper limit, the membranes with low PDMS content performed similarly to commercial glassy polymers, such as polysulfone (PSF) and Matrimid, polycarbonate (PC), for both CO_2_/N_2_ and CO_2_/CH_4_ [34,35,36,37,39]. Our membranes displayed similar CO_2_/N_2_-separation performance compared to most of the copolymer membranes that had PDMS in the rigid polymer, including those that were PI-PDMS-based and PA-PDMS-based (Figure 8a) [29,30]. The membranes also showed comparable CO_2_/N_2_-separation performance to the 6FDA-DABA- and 6FDA-mPDA-based rigid polymer membranes [37]. As shown in Figure 8b, the CO_2_/CH_4_ separation performance of the CE-PDMS-x membranes also demonstrated similar performance to that of PA-PDMS-based copolymer membranes [30].

Nevertheless, the membranes with low PDMS content showed a slightly higher selectivity than the abovementioned glassy polymers because of the enhanced solubility to CO_2_ originating from the ether bonds, together with the selective diffusion cavity provided by crown ether.

In contrast, the membranes with relatively high PDMS content were found to have gas-permeability performances similar to other PDMS-based membranes. The permeability of the polymer membranes synthesized in the present study was predominantly affected by phase separation, especially in the PDMS content range corresponding to 5–10% (i.e., membranes CE-PDMS-5 and CE-PDMS-10). In the range corresponding to CE-PDMS-10 and CE-PDMS-20, there was a critical point for PDMS content beyond which the rigidity of the crown ether started to degrade. The copolymers with PDMS content of 20% or above separated gases similarly to pure PDMS. Copolymers with a lower PDMS content (below 10 wt %) showed excellent selectivities similar to high-performance polyimides.

## 4. Conclusions

In the present study, polyimide was synthesized by copolymerizing crown ether and PDMS, and the gas-permeability performance of the synthesized materials was examined. The crown ether served as a rigid molecule with a high affinity for CO_2_ while providing high thermal stability to the main chains of the polymer and allowing the selective separation of CO_2_. The addition of PDMS, which is a flexible rubbery polymer, not only improved the permeability for all gases but also increased the CO_2_/N_2_ selectivity compared to the pristine 6FDA-CE-PI polymer at a relatively low PDMS content (5 wt %). However, further loading of PDMS (10 wt %) resulted in fine-grained phase separation, adversely affecting the permeability to some extent. In polymers with high PDMS content, it was found to provide a flexible gas-diffusion path, affecting only the permeability without changing the selective gas-separation performance. These results confirmed that the phase-separation phenomenon of the polymer was directly related to the gas separation performance. The copolymers with a PDMS content of 20% or above separated gases like PDMS, whereas those with a lower PDMS content (below 10 wt %) showed excellent selectivities similar to high-performance polyimides. The effectiveness of the newly developed polymers as CO_2_-separation membranes was evaluated, and the performances were similar to commercial polymers. This research opens the possibility of the application of crown ether as a new material for CO_2_-selective gas-separation membranes.

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
