# Peer review of "Development of CO_2_-Selective Polyimide-Based Gas Separation Membranes Using Crown Ether and Polydimethylsiloxane"

_polymers, 2021, doi:10.3390/polym13121927_

Round 1
Reviewer 1 Report
The authors synthesize a serial of CO2-selective polyimides by polymerization of ether with PDMS for gas separation. The work is nicely presented and the result reported is of great interest to readers. However, some part of the manuscript is not well written. I would recommend the publication of the work if the following questions are properly resolved.
- In the Introduction part, the significance of the work is not clearly described. The obstacle to the commercialization of the polymer porous material is the trade-off between permeability and selectivity. How significant of the result report in this work contributed to the challenge.
- In the Method part, l137 to l154 is not organized nicely. Are they NMR data for different polymers? And the description for figures S3 and S4 is not in the final stage. The title for S1 is not complete. References should be provided for some general analysis techniques, such as permeability and selectivity. Such description is valuable for computational studies to accurately model the materials.
- The authors used the general analysis. What new analysis technique could the author propose to rationalize the physics behind the observed behavior?
- The equation number in SI is wrong on diffusivity and permeability. Section S2 is not included in the description of SI.
- In the Results parts, the title of Fig 7 does not include the title for subplots and the logic between subplot a and b is not clear. Fig 7a and 7b are also not clearly described in the main text.
- Why the authors only examine thermal properties for low PDMS contents(fig 3)? What happens for high PDMS contents (over 20% wt loading)?
- l360, similar observations were reported in Ref 26. Could the authors provide some theoretical or simulation works to support the observed behavior?
- In figure 5, the phase separation is observed. Is such structure stable? How about the durability of the material?
- If the PDMS membrane displayed similar separation and a slightly higher selectivity performance compared with commercial polymers. What is the unique advantage of the PDMS membrane compared to commercial polymers?
- Do the data points (red dots) in Fig 8 refer to the data with various PDMS wt% loading?
Some typos:
- l105, the tense is not consistent for this paragraph.
- l425, whitespace is not consistent for this paragraph.
Reviewer 2 Report
The authors describe the synthesis of a set of polyimide-based gas separation membranes using crown ether and PDMS for selective separation of CO2. The materials were characterized using TG, DSC, FTIR, NMR, XRD, AFM, and gas separation. The obtained results are of interest from a practical point of view. However, the MS needs a major revision.
- Title: PDMS -> The use of abbreviations in the MS title is rather inappropriate.
- Figure 1: (a) thinner lines could be used; (b) could be removed into Supplementary Materials (file), cm-1 -> cm-1 (superscripts with -1 should be used).
- Line 225: “In the FTIR spectra, high-intensity absorption peaks” What are the wavenumbers?
- Lines 226-227 “However, in the NMR spectra, another peak was found to appear in the wavelength range corresponding to the benzene rings of 6FDA.” It is unclear.
- Figure 2 may be removed into the SM file.
- Figure 3. (a.u.) -> (arb.un.) because a.u. corresponds to the abbreviation of atomic units.
- The use of the SAXS method (which is more informative for polymers than XRD) could be recommended for detailed textural characterization of the materials. The material morphology could be analyzed in detail using HRTEM.
- The detailed structure of the membranes and its changes due to changes in the composition are rather unclear. Generalized and detailed schemes could be drawn and placed in the SM file.
- On the gas permeability study, it would be interesting to analyze the breakthrough dynamics for the gas mixtures.
- In the SM file, the end should be corrected.
- The MS should be carefully checked and edited by a native speaker. There are a lot of grammar, semantic, and term inaccuracies.
Reviewer 3 Report
This manuscript contains significant amount of characterization. The manuscript is well prepared as well. The reviewer suggests the following comments before further recommendation is given:
- Line 116: What is the meaning of (4) listed in the subtitle 2.2?
- The figures shown in line 135 and 139 need to be labelled as Figure 1 and Figure 2, respectively. Appropriate figure caption needs to be given as well for those two figures.
- The copolymers listed in line 136-154 to be put in a Table. Accordingly, please update the Tables number. In addition, the authors have mistakenly labelled two Tables with the same number “Table 1”.
- Change the “Scheme” to “Figure”. Hence, all Figures number have to be updated accordingly.
- Line 303: The scale of the figures need to be placed.
Round 2
Reviewer 2 Report
The MS could be recommended for publication after minor correction.
Figure 2b. Y-axis “Derivative weight loss”. These derivatives should be negative.
Author Response
Please see the attachment for the reviewer 2's comment and the editor's comment.
